# Design of synthetic collagens that assemble into supramolecular banded fibers as a functional biomaterial testbed

Jinyuan Hu [1,8], Junhui Li [2,8], Jennifer Jiang [3,4], Lingling Wang [1], Jonathan Roth [5], Kenneth N. McGuinness [6,7], Jean Baum [5], Wei Dai [3,4], Yao Sun [2] ✉, Vikas Nanda [6,7] ✉ & Fei Xu [1] ✉

Collagens are the most abundant proteins of the extracellular matrix, and the hierarchical folding and supramolecular assembly of collagens into banded fibers is essential for mediating cell-matrix interactions and tissue mechanics. Collagen extracted from animal tissues is a valuable commodity, but suffers from safety and purity issues, limiting its biomaterials applications. Synthetic collagen biomaterials could address these issues, but their construction requires molecular-level control of folding and supramolecular assembly into ordered banded fibers, comparable to those of natural collagens. Here, we show an innovative class of banded fiber-forming synthetic collagens that recapitulate the morphology and some biological properties of natural collagens. The synthetic collagens comprise a functional-driver module that is flanked by adhesive modules that effectively promote their supramolecular assembly. Multiscale simulations support a plausible molecular-level mechanism of supramolecular assembly, allowing precise design of banded fiber morphology. We also experimentally demonstrate that synthetic fibers stimulate osteoblast differentiation at levels comparable to natural collagen. This work thus deepens understanding of collagen biology and disease by providing a ready source of safe, functional biomaterials that bridge the current gap between the simplicity of peptide biophysical models and the complexity of in vivo animal systems.

Collagens are fibrous proteins. They are the most abundant protein in animal tissues, are a valuable commodity as biomaterials, and are actively involved in biomechanical and biological processes, such as dissipating deformation stress of tissues[1-3], inducing cell polarity during cell–matrix adhesion[4], and directing bone mineralization[5-7]. Collagens have an unusual amino acid composition: rich in glycine (~30%) and proline (~15%)[8], and thus do not follow globular protein folding rules such as hydrophobicity driven core formation, or

[1]Ministry of Education Key Laboratory of Industrial Biotechnology, School of Biotechnology, Jiangnan University, 214122 Wuxi, China. [2]Department of Oral Implantology, School of Stomatology, Tongji University, Shanghai Engineering Research Center of Tooth Restoration and Regeneration, Shanghai, China. [3]Department of Cell Biology and Neuroscience, Rutgers, The State University of New Jersey, Piscataway, NJ 08854, USA. [4]Institute for Quantitative Biomedicine, Rutgers, The State University of New Jersey, Piscataway, NJ 08854, USA. [5]Department of Chemistry and Chemical Biology, Rutgers, The State University of New Jersey, Piscataway, NJ 08854, USA. [6]Department of Biochemistry and Molecular Biology, Robert Wood Johnson Medical School, Rutgers, The State University of New Jersey, New Brunswick, NJ 08901, USA. [7]Center for Advanced Biotechnology and Medicine and the Department of Biochemistry and Molecular Biology, Robert Wood Johnson Medical School, Rutgers, The State University of New Jersey, Piscataway, NJ 08854, USA. [8]These authors contributed equally: Jinyuan Hu, Junhui Li. ✉e-mail: yaosun@tongji.edu.cn; nanda@cabm.rutgers.edu; feixu@jiangnan.edu.cn

presentation of a hydrophilic surface to avoid aggregation[9]. Collagens do not form common secondary structure units like α-helices or β-strands; instead they fold into triple-helices (Fig. 1a), where three polypeptide chains are tethered by a network of main-chain hydrogen bonds, not a hydrophobic core[10,11]. Subsequent to folding, thousands to millions of triple-helices undergo self-assembly to form crystalline supramolecular fibers, providing a foundation for the extracellular matrix[9]. Although collagens are among the most widely applied

biomaterials (e.g., cosmetics, foods, sutures in surgery, and cell-culture substrates in tissue culture), their atypical structural features have hindered understanding of their folding and assembly mechanisms.

Type I collagen exists physiologically as a distinct supramolecular structure of banded fibers with a pattern of alternating dark and light bands that are unmistakable when imaged by an electron microscope[9,12]. This banding originates from an offset, parallel packing

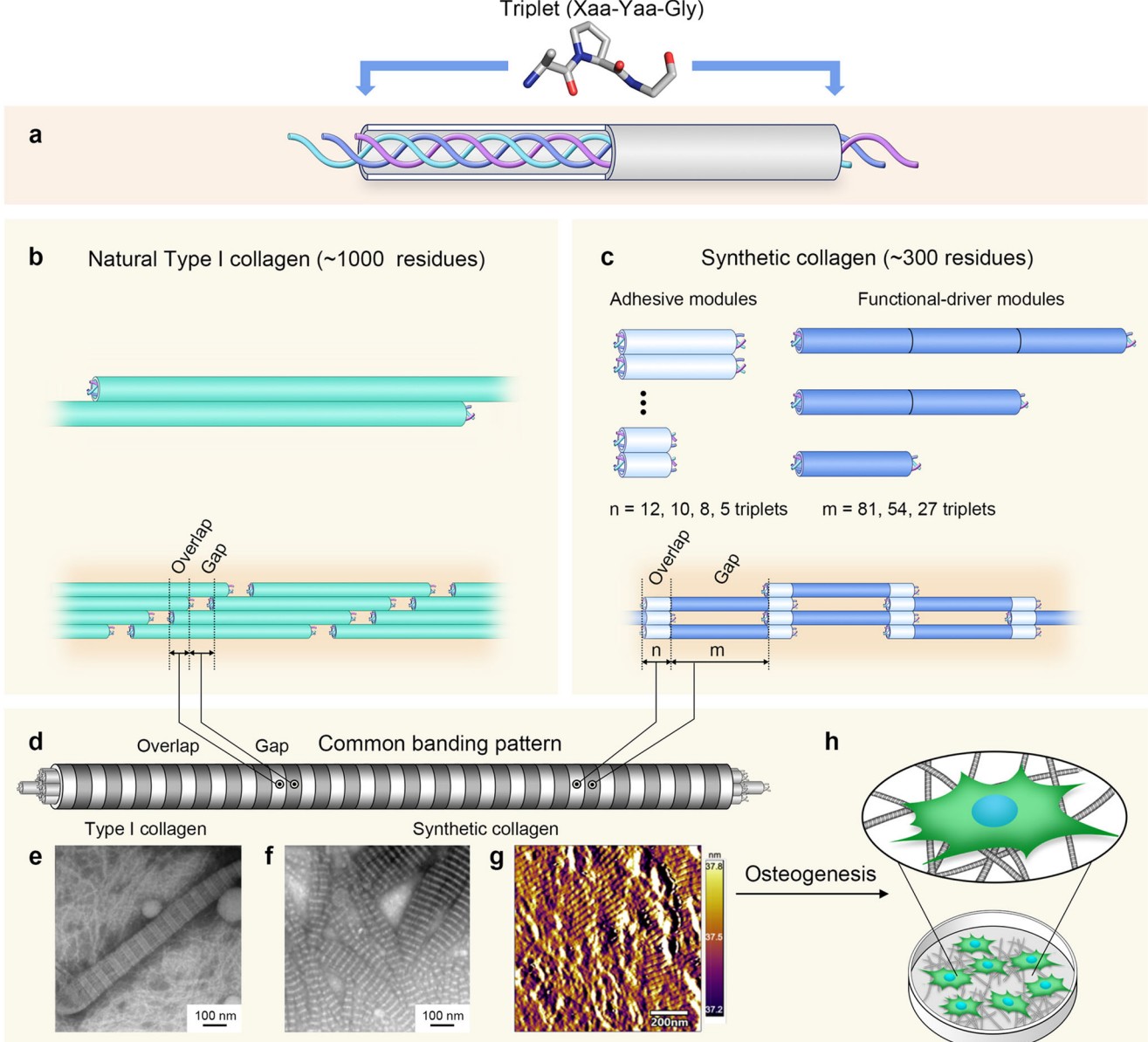

**Fig. 1 | Assembly hierarchy of synthetic collagens built with a modular design strategy vs. natural Type I collagen. a** The folded unit of collagen is the triple-helix, consisting of three supercoiled polypeptide chains with repeating Xaa-Yaa-Gly triplets. **b** In natural Type I collagen, thousands to millions of long thin triple-helices (~1000 residues per chain, diameter ~1 nm, length ~300 nm) are offset and packed in parallel to form fibers[9]. Type I collagen folding and supramolecular assembly have been extensively reviewed elsewhere[60]. **c** Our synthetic collagens were engineered with two adhesives (white) modules flanking a functional-driver (blue) module. Adhesive modules were either hydrophobic, consisting of $n$ Pro-Pro-Gly triplets on both ends of functional-driver modules, or electrostatic, consisting of $n$ Pro-Arg-Gly triplets positioned N-terminal to the functional-driver module, and with the same number of Glu-Pro-Gly triplets at the C-terminal end. Sequences of the functional-driver modules were based on fragments of length $m$ from the Scl2

collagen-like protein of *S. pyogenes*. Type I collagen fibers have a distinct supra-molecular structure, with a pattern of alternating bands of a characteristic periodicity (**d**); these appear as dark and light bands upon negative staining and visualization with transmission electron microscopy (TEM) (**e**). The dark band represents the "gap region" caused by a void between consecutive triple-helices; the light band is the overlap region between neighboring triple-helices. One example, $P_{10}BP_{10}$ ($P_{10}$ refers to 10 PPG triplets, B is the central 27-triplet fragment from Scl2) forms a banded fiber. Periodic bands of $P_{10}BP_{10}$ fibers are visible with TEM (**f**) and atomic force microscopy (AFM) amplitude scans (**g**), experimentally confirming the intended banding morphology. **h** Application testing ultimately showed that our synthetic collagens form banded fibers and strongly promote the osteoblastic adhesion and differentiation of osteoblast precursor cells.

of triple-helices[13–16] (Fig. 1b). The dark band is referred to as the gap region, caused by a void between consecutive triple-helices[12]. The light band is the overlap region between neighboring triple-helices. Natural Type I collagen forms robust banded fibers. It is sourced from animal tissues such as bovine skin or rat tails and is the major type of collagen used in biomedical research and industry[17]. However, animal-sourced Type I collagen suffers from drawbacks such as variable quality and purity, and the risk of prion contamination[17,18]. Synthetic collagens produced under controlled laboratory conditions could address these issues. However, it remains challenging to recapitulate the supramolecular assembly—and thus the banding pattern—of natural collagens[19–22].

Here, we describe a class of synthetic banded fiber collagens, elucidate a detailed mechanism for banded fiber formation, and use these designed materials to experimentally probe the role of banding in promoting osteoblastic differentiation. Individual collagen polypeptide chains are constructed from two components: an adhesive module that promotes supramolecular assembly of folded triple-helices into a banded fiber, and a functional-driver module that can be altered to tailor the material behavior. By varying the lengths and sequences of the two modules, we can precisely specify the widths of the gap and overlap bands. The assembly mechanism is established using both atomistic models and coarse-grained simulations. This class of synthetic collagens advances our physical understanding of collagen supramolecular assembly from the nanometer to micron scale and provides a testbed for studying collagen biology and developing functional biomaterials.

## Results

### Modular design and fiber production

Adhesive modules were constructed from tripeptide repeats designed for electrostatic or hydrophobic inter-helical interactions (Fig. 1c). Hydrophobic adhesive modules were formed from tandem repeats of a motif: (Pro-Pro-Gly)$_n$ sequence, abbreviated as P$_n$. The electrostatic attachment was promoted using a basic domain composed of (Pro-Arg-Gly)$_n$ triplets and an acidic domain composed of (Glu-Pro-Gly)$_n$ triplets on either end of the functional driver, respectively abbreviated R$_n$ and E$_n$. In the case of electrostatic adhesive modules, the total length was maintained at 15 residues with the remainder of the module composed of Pro-Pro-Gly triplets (Fig. 1c).

The functional-driver module is not responsible for assembly, and by design should be a largely unconstrained repeated Xaa-Yaa-Gly triplet sequence that can be tailored for specific applications. We explored the plasticity of the functional-driver sequence using fragments of *Streptococcus pyogenes* Scl2, a collagen-like bacterial protein with a triplet insertion[23]. Three fragments, referred to as A–C, were chosen for their similar lengths but different amino acid compositions and net charges[24–26]. Functional-driver modules constructed from combinations of fragments with varied compositions and lengths were inserted between two adhesive modules (Fig. 1c). Our nomenclature for the synthetic collagens is based on the arrangement of modules: for example, we use "R$_5$BE$_5$" for a variable domain constructed from the Scl2 B fragment flanked by electrostatic attachment modules and use "P$_{10}$BBP$_{10}$" for two tandem Scl2 B fragments flanked by ten Pro-Pro-Gly triplets. Complete sequences of all constructs are in Supplementary Table 1.

The folding and assembly protocol for the designed synthetic collagens parallels many aspects of natural collagen assembly. Type I collagen is expressed in the cell in a procollagen form, where the triple-helix is flanked by globular domains that assist in folding; these are subsequently cleaved prior to supramolecular assembly of banded fibers[27]. Bacterial Scl2 also has a globular domain (V-domain) that facilitates folding and expression[24]. We included the V-domain prior to the first adhesive module, separated by a trypsin cleavage site (Supplementary Table 1). Synthetic collagens folded into triple-helices

within *Escherichia coli* cells. After expression and purification, the synthetic collagen was digested with trypsin to remove the V-domain[28] and incubated at 4 °C for several days, allowing for supramolecular assembly into banded fibers. Details of protein production and validation are described in the "Methods" section and Supporting Information (Supplementary Figs. 1–3).

### Controlling fiber morphology

According to the design scheme, the number of amino acids in the adhesive module should specify the length of the overlap region, and the functional-driver module size should determine the length of the gap. The first design studied—P$_{10}$BP$_{10}$—formed fibers with periodic light and dark bands, as observed in negative staining electron microscopy images (Fig. 1f). Assembly of banded fibers was robust, and reproducible multiple times under a variety of solution conditions including 50–150 mM NaCl concentrations, pH 3–11, incubation times from 0.5 to 70 days, and temperatures from 4 to 37 °C (Supplementary Figs. 4–6). Although P$_{10}$BP$_{10}$ assembles into fibers, micro-rheology experiments show fibers do not form a viscoelastic gel either at a low (1 mg mL$^{-1}$) or high (5 mg mL$^{-1}$) concentration (Supplementary Fig. 7).

In a triple-helix, each residue spans 0.29 nm along the super-helical axis, allowing one to estimate the size of a domain based on the length of its sequence. Band widths were ~10 nm (light) and 24.9 nm (dark), matching the expected sizes of the P$_{10}$ (30 residues) and B (81 residues) sequences, respectively (Fig. 3c). Surface contour maps of P$_{10}$BP$_{10}$ banded fibers imaged by atomic force microscopy (AFM) showed the lighter bands were ~1 nm high, matching the diameter of a triple-helix (Fig. 1g, Supplementary Fig. 8). Combined, TEM and AFM support the design approach where the light bands (the overlap region) in P$_{10}$BP$_{10}$ are formed by adhesive modules, and the dark bands (the gap region) are formed by the functional-driver modules.

To investigate the three-dimensional morphology of synthetic collagen fibers, we generated P$_{10}$BP$_{10}$ and P$_{10}$BBP$_{10}$ fibers under the same conditions used for TEM and imaged the samples using a Talos Arctica cryo-electron microscope. Tomograms of P$_{10}$BP$_{10}$ fibers showed that collagen fibers form 3D bundles that are 500 – 800 nm in length and taper at the ends. These bundles exhibited a distinct 3D structure and periodic light and dark bands (Fig. 2a-c, Supplementary Mov. 1). Measurements of light bands (23.6 nm) and dark bands (9.5 nm) were consistent with dimensions of gap and overlap band-widths respectively, but of opposite intensity as compared to TEM. Using Fourier transforms, we calculated the average periodicity to be 32.4 nm (Fig. 2d), consistent with measurements in real space (33.2 nm, Fig. 2b) and in AFM (32.7 nm, Supplementary Fig. 8). To examine the domain organization, we performed volume rendering of the P$_{10}$BP$_{10}$ fiber tomograms using UCSF Chimera and observed that collagen fibers within the bundles show a pattern of dense overlap zones and sparse gap zones (Fig. 2e, Supplementary Mov. 1). This observation further supports that the overlap region (dense zone) is formed by tightly stacked adhesive modules and that the gap region (sparse zone) is formed by loosely stacked functional-driver modules. Tomograms of P$_{10}$BBP$_{10}$ fibers revealed similar 3D fiber bundles and a characteristic periodic light and dark banding pattern with a 55.6 nm periodicity, consistent with observations in TEM (Supplementary Fig. 23, Supplementary Mov. 2). The light band widths of P$_{10}$BBP$_{10}$ were twice of that of P$_{10}$BP$_{10}$ fibers. The banded fibers formed by this class of synthetic collagens were also three-dimensional.

We tested our design approach by preventing supramolecular banded fiber assembly by several strategies: denaturing the triple-helix structure or altering the order of modules. The P$_{10}$BP$_{10}$ triple-helix unfolds with a transition temperature ($T_m$) of 52 °C (measured by CD—Supplementary Fig. 9j–l), and 55.2 °C measured by differential scanning calorimetry (DSC) (Supplementary Fig. 10c). Heating P$_{10}$BP$_{10}$ above the $T_m$ prior to assembly prevented banded fiber formation (Supplementary Fig. 11). Synthetic collagens with alternate module

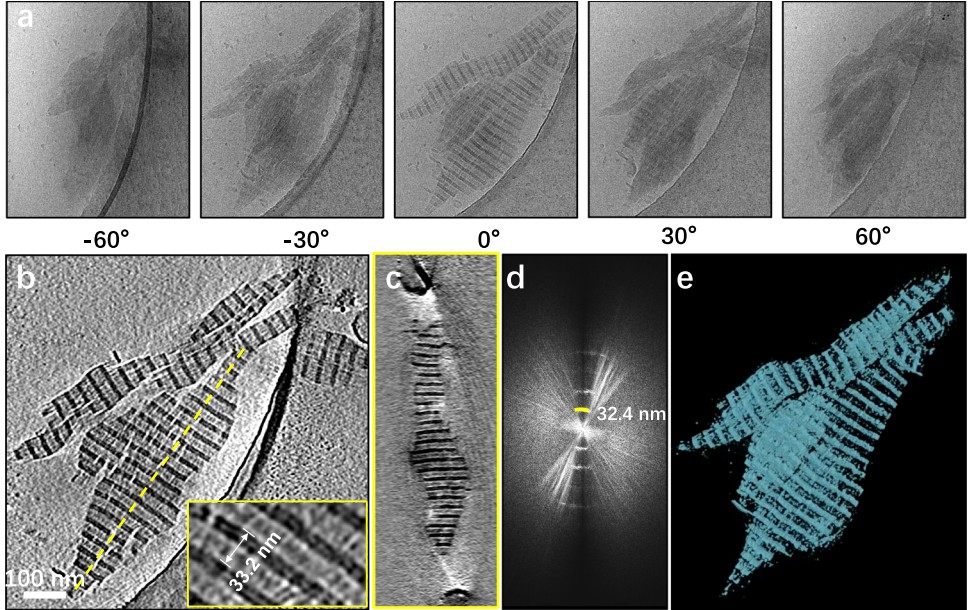

**Fig. 2 | Cryo-electron tomography of P10BP10 collagen fibrils. a** Tilt series of collagen fibers at tilt angles ranging from −60° to +60°. **b** Slice views of the tomogram of $P_{10}BP_{10}$ collagen fibers with characteristic banding patterns. Inset in **b** shows a zoomed-in view with measurement of periodicity. **c** Cross-section of a collagen bundle as indicated by a dashed yellow line in **b. d** Fourier transform of the image in **b** showing banding periodicity. **e** Volume rendering of collagen fibers showing dense overlap and sparse gap zones.

topologies−i.e. containing one adhesive module before, in the middle of, or after the functional-driver B module ($P_{10}$B, ½$BP_{10}$½B and $BP_{10}$)− folded into triple-helices (Supplementary Fig. 9a−c), but did not assemble into banded fibers. Thus, a folded triple-helix and a topology of two adhesive modules flanking a functional-driver are prerequisites for synthetic collagens that form banded fibers.

We show that one can independently target the properties of dark and light bands by varying the functional-driver and adhesive modules, respectively. The Scl2 B fragment was incorporated into $P_nBP_n$ and $R_5BE_5$ designs, producing banded fibers where the dark band consisted of a width of around 24 nm, matching the calculated length of an 81-residue triple-helix. In our design, the composition of the functional-driver module is relatively unconstrained as long as it folds into a triple-helical structure. To test this, we used Scl2 fragments A and C as functional-driver modules that are of similar size to B, but their amino acid compositions are unique, with A having very few charges, and C rich in charged residues. All three designs, $P_{10}AP_{10}$, $P_{10}BP_{10}$, and $P_{10}CP_{10}$ readily assembled into fibers, with similar bandwidths (Fig. 3a−c, e, h and Supplementary Figs. 12, 13). Thus, the functional-driver module sequence can be modified without noticeably perturbing banded fiber morphology.

It should be possible to increase the width of the dark band by lengthening the functional driver. $P_{10}BBP_{10}$ and $P_{10}ABCP_{10}$ folded into triple-helices (Supplementary Fig. 9j−l, 10) and assembled into banded fibers. The dark bandwidths of $P_{10}BBP_{10}$ and $P_{10}ABCP_{10}$ were twice and three times that of $P_{10}BP_{10}$, while the light bandwidths were unchanged (Fig. 3e−h and Supplementary Fig. 13). Indeed, the 59 nm dark bandwidth of $P_{10}BBP_{10}$ is similar that of the Type I collagen 67 nm D-period[9,12], showing one can engineer banded fibers with biologically relevant dimensions.

## Molecular mechanism of adhesion

We explored two strategies of triple-helix adhesion based on modules composed of amino acids that make hydrophobic or electrostatic interactions. Briefly, we anticipated that the strengths of these interactions would scale with the size of the interface. To assess the size and energetic requirements for hydrophobically-driven assembly,

we constructed a series of synthetic collagens of the form $P_nBP_n$, of varying lengths $n = 5, 8, 10$, and 12. All designs folded into triple-helices as measured by CD, with $T_m$ values higher than the parent Scl2 domain[28] (Supplementary Fig. 9d−f). Aggregation, but not banding, was observed for the $P_5$ adhesive module in $P_5BP_5$ and $P_5BBP_5$ (Supplementary Fig. 14). When $n \geq 8$, all $P_nBP_n$ designs formed banded fibers (Fig. 3j−l and Supplementary Fig. 15), where the light bandwidth was directly proportional to $n$, confirming that the light band is formed by the adhesive modules.

Eight repeats of a Pro-Pro-Gly triplet were sufficient to promote assembly, suggesting an interaction threshold for adhesion-dependent on the size of the interface. To quantify this potential requirement, we computed interactions between pairs of triple-helical adhesive modules using the protCAD software platform[29]; note that protCAD was developed for structure-guided protein engineering, and enables sampling of intramolecular torsional rotations as well as intermolecular degrees of freedom. Interaction energies were scored using the AMBER ff14sb molecular mechanics force field[30]. Interhelical packing was optimized by sampling rigid body translational and rotational parameters[31]. In the optimized conformation, an extensive packing interface was generated between triple-helical ridges formed by the proline sidechains (Fig. 4a). As the module length, $n$, increased from 5 to 12 triplets, the calculated interaction energy score increased linearly from −30 to −70 kcal mol⁻¹ (Fig. 4d and Supplementary Table 2). Based on these calculations, the interaction threshold for supramolecular assembly would lie somewhere between −30 and −50 kcal mol⁻¹.

Electrostatic adhesion-driven assembly was also investigated. $R_5BE_5$ folded into triple-helices and assembled into banded fibers. The light band had a mean width of 5.8 nm, matching the expected dimensions of a 15-residue sequence. The dark bandwidths were consistent with that of Scl2 fragment B, unaltered by switching from hydrophobic to electrostatic adhesive domains (Fig. 3i−l and Supplementary Fig. 15). Designs with shorter electrostatic adhesive modules− $R_3BE_3$ and $R_1BE_1$−did not assemble into banded fibers. This further supports the plausibility of an energetic interaction threshold for the electrostatic-driven supramolecular assembly.

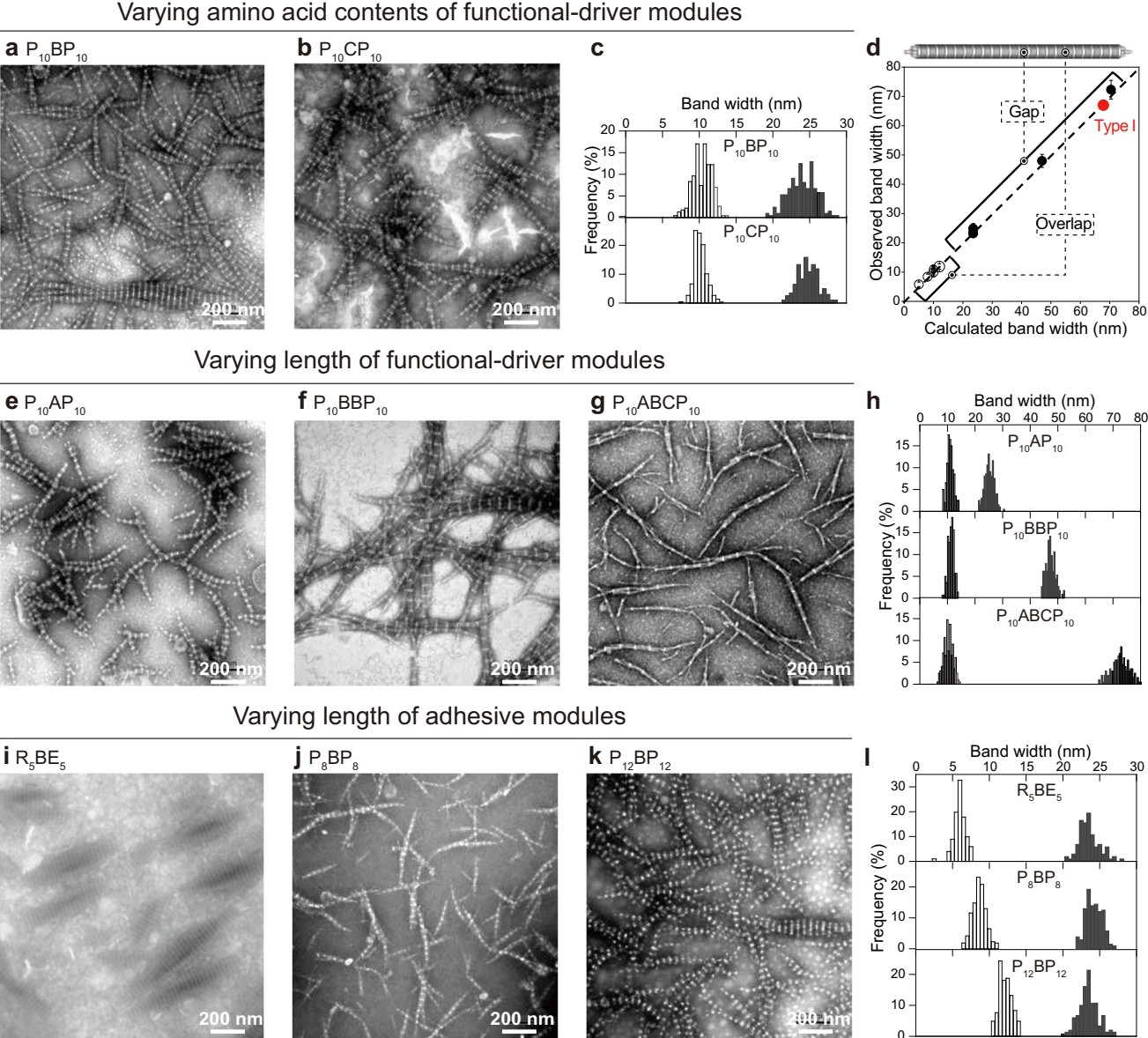

**Fig. 3 | Fine-tuning width of dark and light bands by exchanging functional driver and adhesive modules. a** and **b** The width of the dark bands remained similar when the functional-driver modules had the same length yet had differing amino acid compositions in $P_{10}BP_{10}$ (**a**) and $P_{10}CP_{10}$ (**b**); the bandwidth distribution is plotted in **c**. The width of the dark bands increased as the functional-driver modules increased from 27 to 81 triplets in $P_{10}AP_{10}$ (**e**), $P_{10}BBP_{10}$ (**f**), and $P_{10}ABCP_{10}$ (**g**); the width distribution is plotted in **h**. The width of the light bands increased as the adhesive modules increased from 5 to 12 triplets in $R_5BE_5$ (**i**), $P_8BP_8$ (**j**), and $P_{12}BP_{12}$ (**k**); the width distribution of both the light and dark bands are plotted in **l**. **d** Observed bandwidths were plotted against those calculated from module length, multiplied by the average helical rise (-2.9 Å) per residue[61]. The 67 nm Type I collagen D-period assumes a repeat length of 234 residues[32]. The dashed lines indicated that the light and dark bands of the synthetic collagen, respectively, correspond to the overlap and gap region of Type I collagen. Solutions of the designed collagens at 0.5 mM in 10 mM phosphate buffer (pH 7.0) were incubated at 4 °C for 3 days, then negatively stained, and visualized with Transmission Electron Microscopy (TEM) (bandwidths were measured using ImageJ, $n = 200$). Images are all at the same magnification.

We again used protCAD to compute the interaction energy scores, this time with atomic models of electrostatic adhesive modules. In addition to inter-helical degrees of freedom, we also sampled side-chain rotameric states of the Arg and Glu amino acids. Electrostatic modules tended to associate at larger inter-helical distances than hydrophobic ones (Fig. 4a, b), likely accommodating the larger Arg and Glu sidechains at the interface. Decreased contributions from *van der Waals* packing forces due to greater inter-helical separations were compensated by electrostatic interactions. In the $R_5 + E_5$ calculations, Arg and Glu sidechains interdigitated, forming multiple strong (-3.3 Å) interhelical salt bridges (Fig. 4b–c). The interaction score was proportional to the number of charged residues in each adhesive module. Given that $R_3BE_3$ and $R_1BE_1$ did not form banded fibers, an interaction threshold for assembly between −30 and −50 kcal mol⁻¹ apparently exists for electrostatically driven adhesion. This is the same range as for hydrophobic interactions, suggesting a common interaction threshold for supramolecular assembly across all designs.

It was thought that the functional-driver sequences derived from Scl2 might coincidentally contain regions that promote interhelical associations through hydrophobic or electrostatic interactions, which could confound our proposed mechanism for assembly. As atomistic calculations are impractical for evaluating all possible supramolecular interactions at this scale, we applied a discrete, primary sequence-based approach used previously to model natural collagen interactions[20,32]. Briefly, two sequences are aligned in a series of poses by shifting one amino acid at a time (Supplementary Fig. 16a).

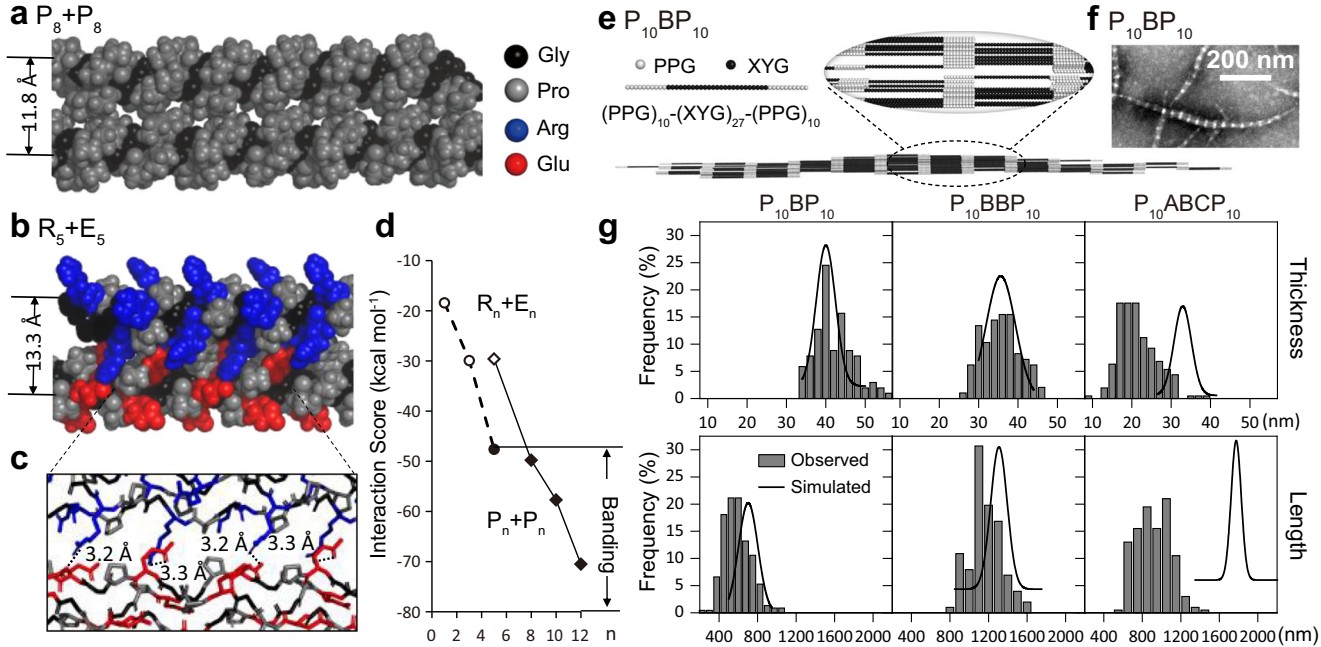

**Fig. 4 | Multi-scale computational simulations to understand the assembly mechanism of the banded fibers. a–d** Atomic-level models of the interactions between adhesive modules were built with the software package ProtCAD[29], with conformations optimized to achieve the lowest interaction score based on simulated annealing. Structural models of $P_8 + P_8$ (**a**) and $R_5 + E_5$ (**b**), with the lowest interaction scores shown as space-filling models. Predicted salt bridges in $R_5 + E_5$ are shown with dashed lines in **c**. Predicted interaction scores of $P_n + P_n$ (diamonds) and $R_n + E_n$ (circles) were plotted against the number, n, of the triplets in the adhesive modules in **d**. Filled data points represent designs for which banded fibers were observed. **e** Fiber morphology and dimensions were simulated with a diffusion-limited aggregation simulation[35]. A triple-helical unit, $[(Xaa\text{-}Yaa\text{-}Gly)_1]_3$,

was simplified as a sphere. The triple-helices of $P_{10}BP_{10}$ were simplified as a rod with adhesive ends (gray) and central functional-driver regions (black). The assembly process was simulated by allowing 1200 rods to diffuse randomly along a hexagonal lattice, until contacting an existing rod through their adhesive ends. In the simulated fiber, the adhesive modules formed regions with a high density of spheres, corresponding to the light bands under TEM as shown in **f**, while the functional-driver modules formed low-density regions, corresponding to the dark bands. **g** As the length of the functional-driver modules increased in $P_{10}BP_{10}$, $P_{10}BBP_{10}$, and $P_{10}ABCP_{10}$, the fiber dimensions—including the mean values for thickness (upper panel) and length (lower panel)—were obtained from DLA simulations and were co-plotted with those in the TEM images (analyzed using ImageJ).

Interaction scores are calculated based on the number of hydrophobic and electrostatic contacts, allowing us to rank the different poses. As expected, when this calculation was applied to Type I collagen, the highest interaction scores occur at staggers every 231–236 residues (Supplementary Fig. 17), consistent with the length of D-spacing[9,12]. For synthetic collagens $P_{10}BP_{10}$, $P_{10}BBP_{10}$, and $P_{10}ABCP_{10}$, top-scoring poses comprised fully overlapping adhesive domains (Supplementary Fig. 16b, c). Therefore, interactions between the proline-rich adhesive modules, rather than incidental interactions between functional-driver modules, likely drive assembly.

**Simulating supramolecular assembly**

The multiscale structural features of natural collagens, from the atomic to the macroscopic, all contribute to biological activity[27,33]. When assembled in vitro—as would be done in tissue culture for example—animal collagens will form long banded fibers that are thicker in the middle and tapered at the ends[34]. This is consistent with an assembly pathway known as diffusion-limited aggregation (DLA)[35], where individual triple-helices attach to a growing fiber at any point along its length (Supplementary Fig. 18). Micrographs of our synthetic collagens showed this anticipated morphology, indicating DLA as the likely operating mechanism of supramolecular assembly. It would be useful to predict supramolecular features on the micron scale for synthetic collagen assemblies composed of thousands or more triple-helices. As atomistic methods are computationally expensive at such scales, we used a minimalistic coarse-grained Monte Carlo DLA model, developed previously with Type I collagen[35] and synthetic peptides[36], to predict supramolecular features on the ten to hundred-micron scale.

DLA simulations were performed using coarse-grained models of synthetic collagens, where individual triple-helices were approximated as rigid rods composed of adhesive and functional driver modules (Fig. 4e, f). DLA simulations produced banded fibers consistent with the ones observed through TEM and cryoET: these were thicker in the middle and tapered at the ends. As expected, the coarse-grained models showed that the gap region dark bands formed by functional-driver modules had a lower packing density than the overlap regions (light bands) formed by the adhesive domains (Supplementary Fig. 19). These spaces would allow a negative TEM stain to infiltrate the gap regions and produce a darker color. Conversely, overlapping attachment domains had a higher density in DLA models, consistent with light bands resisting stain.

Models from DLA simulations were also compared to observed fiber thicknesses and lengths. Simulated fibers were generated for $P_{10}BP_{10}$ containing 50–3000 individual triple-helices (Supplementary Fig. 20). Simulated dimensions best-matched observations for fibers composed of ~1200 rods (Fig. 4g). The length and thickness of the simulated fibers increased with the number of rods (Supplementary Fig. 20). However, the aspect ratio (length/thickness) decreased only slightly, still close to the values extracted from TEM (Supplementary Fig. 21), consistent with previous diffusion models of collagen[37]. Simulated dimensions best-matched observations for fibers composed of ~1200 rods (Fig. 4g). These observations held for the longer $P_{10}BBP_{10}$ constructs. Only for the longest synthetic collagen, $P_{10}ABCP_{10}$, did the simulations significantly diverge from observations. This may be due to a number of factors, including potential deviations from a rigid rod approximation for long designs that would increase the anisotropy of diffusion[38]. It would be useful to explore alternate models of collagen

that incorporate triple-helical flexibility[39]. These findings, including the edge case with $P_{10}ABCP_{10}$, demonstrate how coarse-grained modeling tools enhance our physical understanding of the collagen assembly process. In establishing DLA as the pathway from triple-helices to banded fibers, we can computationally sample module configurations, prior to studying them in the laboratory, showcasing the utility of synthetic collagens as a testbed for developing designer biomaterials.

## Banded fibers support osteoblastic differentiation

Type I collagen is the primary extracellular matrix protein in bone, and its assembly into a banded fiber is known to be essential for osteogenesis and matrix mineralization[5,6,40]. We, therefore, tested whether the cell culture of osteoblast precursor MC3T3-E1 cells on synthetic collagens that form banded fibers would significantly promote cell adhesion and differentiation over related, non-banded protein substrates. Cell behaviors on synthetic banded fiber collagen substrates ($P_{10}BP_{10}$ and $P_{10}BBP_{10}$) were compared to related fiber-incompetent substrates (B and $P_5BP_5$). Type I collagen and bovine serum albumin (BSA) were included as positive and negative controls, respectively. Our results ultimately showed that cultured osteoblasts preferentially thrived and differentiated on synthetic banded collagen substrates.

Osteoblast viabilities and proliferation rates for all substrates were comparable (Fig. 5a), confirming that our synthetic collagens are biocompatible. Synthetic collagen banded fiber and Type I collagen substrates promoted higher cell adhesion than the banding-incompetent substrates and BSA control (Fig. 5b). Natural collagen could promote the formation of focal adhesion sites and recruitment of cytoplasmic actin-binding proteins (e.g., vinculin and α-actinin)[41,42]. We measured *VCL* and *ACTN* mRNA levels, and found that the presence of the banded fiber substrates promoted the transcription of both genes (Fig. 5c). Cytoskeletal phenotypes in response to cell adhesion were evident when we assayed cell spreading area and performed cytoskeletal staining with phalloidin (Fig. 5d, e). The cells which were adhered to the banded fiber substrates had larger spreading areas with more obvious cytoskeleton extension, and had prominent cell dendrites, indicating accelerated maturation of such cells (Fig. 5d, e).

The banded fiber substrates also promoted MC3T3-E1 cell differentiation into mature osteoblasts. Alkaline phosphatase (ALP) is an extracellular bone enzyme that increases the local concentration of inorganic phosphate and facilitates mineralization[43], and is a typical biomarker of osteoblastic differentiation and incipient mineralization. ALP-stained areas for $P_{10}BP_{10}$ and $P_{10}BBP_{10}$ were comparable to that of Type I collagen and significantly larger than for the B, $P_5BP_5$, or BSA sample groups (Fig. 5f, g). The expression of *RUNX2*, an osteoblastic-specific transcription factor[44], was induced by the synthetic banded fiber collagens $P_{10}BP_{10}$ at levels higher than that of the banding-incompetent substrate, BSA control, and even natural Type I collagen (Fig. 5h).

The similar outcomes of cells on $P_{10}BP_{10}$ and $P_{10}BBP_{10}$ suggest that common features of these two designs and that of Type I collagen stimulate differentiation. We hypothesize that the banding of the synthetic collagen fibers aligns the engineered integrin binding sites (GFPGER) in the fragment B[45] (see sequences in Supplementary Table 1), resulting in clustering of cell receptors, promoting and downstream signaling (Supplementary Fig. 24)[46].

Synthetic collagens that form banded fibers enhanced cell compatibility of MC3T3-E1 osteoblast precursor cells, and promoted the expression of osteoblastic differentiation markers *ALP* and *RUNX2* in these cells, consistent with observations that the supramolecular structure of natural collagens is actively involved in osteogenesis and bone mineralization[5,6,40]. Previous studies have reported that collagen from animal models with osteogenesis-relevant diseases (e.g., osteogenesis imperfecta (OI) and osteopenia) display aberrant Type I collagen banding[47–49]. Reported issues from the industry regarding the purity and quality of animal-extracted collagens also limit both the

precision of data and the confidence in results from experiments addressing the pathogenic impacts of aberrant collagen structures. In addition, by engineering disease-causing mutations into collagen modules designed based on hypotheses from clinical or biological insights, synthetic collagens can be further used to investigate the molecular mechanisms of cell–matrix interactions in osteogenesis, as well as pathogenesis-related mechanisms in great detail.

## Discussion

We designed, modeled, and tested an innovative class of synthetic collagens comprising an adhesive module to promote supramolecular assembly into classic collagen banded fibers, and a functional-driver module that can be precisely tailored to obtain desired material behaviors. The robust expression and assembly of these banded fibers for a range of materials with different combinations of modules can provide pure, safe, and functionally designable collagen biomaterials. Our demonstration that $P_{10}BP_{10}$ and $P_{10}BBP_{10}$ could promote the differentiation of MC3T3-E1 osteoblast precursor cells at levels comparable to natural collagen confirms that these banded fibers are valuable as biomedical research tools and as a testbed for designer functional biomaterials. Although the designed collagens did not form hydrogels, they still can be potentially used as coating materials on metal implants[50]. It may be possible to design functional drivers that promote the formation of a supramolecular viscoelastic gel.

Computational models that accurately recapitulate the supramolecular assembly process for these fibrous proteins represent a major conceptual advance to better define the rules governing the supramolecular assembly of natural collagen. Specifically, we have established energetic interaction thresholds between adhesive modules and have set up a coarse-grained DLA model to help understand how protein length and composition dictate the morphology of assembled fibers. Notably, we show that extended proline-rich sequences can be used to modulate the strength of intra-fiber adhesion, and changing the lengths of adhesive and functional-driver modules can vary the period of fiber banding at biologically relevant scales.

Functionally designable synthetic collagens with tunable banding morphologies provide an unprecedented tool for exploring collagen–collagen and collagen–cell interactions in biology and disease. To date, investigations of collagen diseases such as OI have largely depended on animal models, where the specific role of collagen cannot be studied in isolation[47–49], or on collagen peptides and fragments[33], which do not recapitulate the supramolecular nature of natural collagen fibers. These designs provide a supramolecular platform for hosting OI mutations of interest, enabling characterization of their precise impacts in disrupting collagen fiber assembly and during osteogenesis. Our synthetic banded fibers provide a flexible platform that fills the gap between simplified biophysical models of collagen-like peptides and complex in vivo systems of model animals and can foster discoveries about collagen-involved biological and pathological processes.

## Methods

### Plasmid construction, protein expression, and purification

Synthetic genes of the designed collagens (Supplementary Table 1), which contained a 5′ (*Nco*I) and 5′UTR ([*GC*]) and 3′ (*Bam*HI), were optimized for the *E. coli* expression system. For each designed collagen, its gene was cloned into the pColdIII-Tu vector through *Nco*I and *Bam*HI. The pColdIII-Tu vector was obtained from pColdIII site-directed mutation of the *Nde*I to *Nco*I by primer S1 (CTCGAGG GATCCGAATTCA) and A1 (GAGCTCCATGGGCACTTTG). All verified constructs were transformed into *E. coli* BL21 (DE3) (Supplementary Fig. 1). The primary seed culture (1 mL) was used to inoculate 100 mL of TB medium containing 100 μg mL$^{-1}$ ampicillin at 37 °C cultured for 24 h. The bacteria liquid was cooled to 25 °C and induced with 1 mM

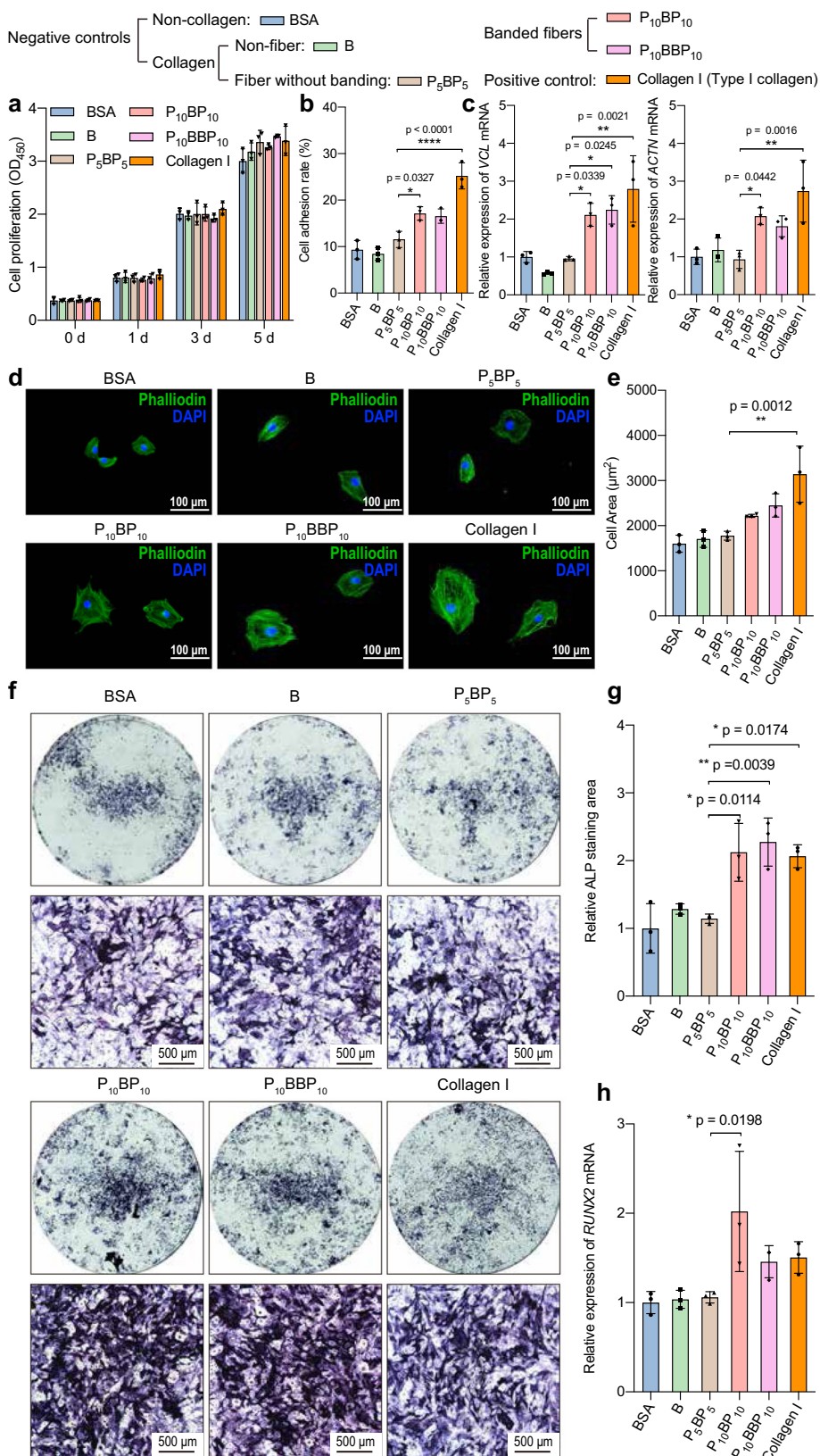

IPTG. After 10 h incubation, cells were further cooled to 15 °C for 14 h[51]. Cells were harvested by centrifugation at 4 °C. The cell pellets were resuspended in binding buffer (20 mM NaPO₄, 500 mM NaCl, 10 mM imidazole, pH 7.4), lysed with sonication, centrifugated at 10,000×*g* for 20 min at 4 °C and filtered with the microporous membrane (0.45 µm) to remove the cell debris. The designed collagen with a His₆

tag at N-terminus was purified with 5 mL HisTrap HP column using the stepwise increasing concentrations of imidazole (130, 160, and 400 mM). The globular folding domain, V-domain, of the designed collagen was removed by incubating with trypsin at 1:1000 (w:w) ratio at 25 °C overnight, which was dialyzed and freeze-dried. As the trypsin is autolyzed into fragments within several hours[52], the fragments were

**Fig. 5 | Proliferation, adhesion, and osteoblastic differentiation of MC3T3-E1 osteoblast precursor cells induced by the banded fibers of synthetic collagens.** **a**–**e** A non-osteoblastic inducing cell culture was seeded on plates coated with several types of synthetic collagen samples as labeled at the top of the figure. Cell viability at various time points after seeding (0, 1, 3, or 5 days) was analyzed with CCK-8 assays (**a**), In order to confirm cell adhesion at the transcription level, cell adhesion rates were evaluated (**b**), and the relative mRNA expression of the adhesion-related genes *VCL* and *ACTN* were measured using qPCR after seeding for 6 h (**c**). **d** The cells were stained with Phalloidin staining and visualized with fluorescence microscopy (scale bar = 100 μm); **e** stained areas of the adhered cells were calculated using ImageJ. Osteoblastic-differentiation tests were carried out on plates coated with the same collagen samples as in **a**–**e**. The experiments were carried out in a 24-well cell-culture plate for 7 days. **f**, **g** Representative alkaline phosphatase (ALP) staining images of a whole well (upper panels) and the central region of the well (scale bar 500 μm, lower panels) were taken under a light microscope, and the ALP-positive area of the whole well was calculated. **h** In order to test the osteoblastic differentiation at the transcription level, the relative mRNA expression of the osteoblastic-differentiation-related gene *RUNX2* was measured with qPCR. All of the experiments had three replicates, and the data are represented as the mean ± s.d. (*n* = 3). The significance of differences among groups was assessed using ordinary one-way ANOVA with multiple comparisons test.

eliminated from the protein solutions with the dialysis. The molecular weights and purity (>95%) of the designed collagens were validated by SDS−PAGE and MALDI-TOF mass spectrometry (Supplementary Figs. 2 and 3).

## Circular dichroism (CD) spectroscopy

CD Spectra were obtained via an applied photophysics chirascan with a Peltier temperature controller, using quartz cuvettes with 1 mm path length (Model 110-OS, Hellma USA). Protein solutions (1 mg mL⁻¹ in 10 mM PB buffer) were equilibrated at 4 °C for ~24 h before measurements. Wavelength scans were collected at 4 °C from 190 to 260 nm (step 1 nm, averaging time 5 s). For temperature transition experiments from 4 to 80 °C, the ellipticity was monitored at 220 nm at an average heating rate of 1 °C for 6 min⁻¹ with 8 s equilibration time. The first derivative of the melting curves was used to obtain the melting temperatures of collagen samples.

## Differential scanning calorimetry (DSC)

DSC was carried out for the collagen proteins in a NANO DSC instrument (TA Instruments) coupled with a thermal data analysis system, Nano-Analyze software. The collagen solution was at 5 mg/mL in 10 mM phosphate buffer at pH 7 and incubated at 4 °C for 3 days before measurements. The 10 mM phosphate buffer was used for the baseline scan at least three times. Sample solutions were loaded at 20 °C into the cell and heated at a rate of 1 °C/min.

## MALDI-TOF mass spectrometry

MALDI-TOF was used to confirm the molecular mass of the designed collagen (Bruker Daltonics UltrafleXtreme, USA). 1 μL of 1 mg mL⁻¹ sample was dropped in a 600 μm AnchorChip target plate (Bruker) and allowed to dry, using 5 mg mL⁻¹ 2,5-dihydroxyacetophenone, 25% (v/v) ethanol, 1.5 mg mL⁻¹ diammonium hydrogen citrate as the matrix in linear mode, BSA was used for external calibration. Or using 10 mg mL⁻¹ sinapinic acid in 50% acetonitrile, 0.1% TFA as the matrix. 1 μL of the sample was applied onto a plate with apo-myoglobin as the internal standard. Data was acquired with AB Sciex 4800 at positive linear mid-mass mode.

## Transmission electron microscopy (TEM)

Solutions of the designed collagens at 0.5 mM in 10 mM phosphate buffer (pH 7.0) were incubated at 4 °C for 3 days. The solution was placed onto a copper grid and was incubated for 45 s, after which the excess solvent was absorbed from the grid using filter paper. The sample was negatively stained with 0.75% phosphotungstic acid for 20 s, and the excess stain was absorbed away and imaged with Hitachi H-7650 (Hitachi, Japan) Electron Microscope. The light- and dark-band widths were measured with ImageJ about 200 times from at least 5 TEM images.

## Atomic force microscope (AFM)

Tapping mode was performed on a Cypher ES AFM (Asylum Research, Oxford Instruments) at room temperature. AC240 silicon nitride cantilevers sourced from Oxford Instruments were used for imaging in the air. These cantilevers have a nominal spring constant of kc = 2.0 N m⁻¹, and radii of *r* = 7 nm. Sample solutions were deposited onto freshly cleaved mica, allowed 5 min for binding, and then washed with 2 mL of water. Samples were then allowed to air dry for 1 h in a laminar flow cabinet before being placed in the AFM for imaging and analyzed using Asylum Research software. Before the Fourier transform was used to analyze the banded fibers in AFM images, EMD denoising protocol was used to remove the baseline drift and noise[53]. The data processing was carried out with Matlab_R2016a with a source code, which was uploaded to GitHub (https://github.com/JinyuanHu/FFT_AFM.git)

## Microrheology

Solutions of the designed collagens and Type I collagen at 1 and 5 mg mL⁻¹ in 10 mM phosphate buffer (pH 7.0) were incubated at 4 °C for 3 days. Then Young's moduli, *G′* and *G″*, were determined as previously described[54]. Fluorescent polystyrene beads were added to the solution and record the trajectories of the fluorescent beads were by fluorescence microscopy, then determine the mean square displacement as a function of time and Young's moduli, *G′* and *G″*, were determined from the Generalized Stokes−Einstein relationship.

## Calculation of inter-triple helical interactions

The interaction energy scores of the designed collagen were computed according to Hulmes et al. using a sequence-based model[32]. Charged pair (K/R, D/E) and hydrophobic pair (V, M, I, L, F, P) were given a score of one. Repulsion pairs were given a score of negative ones. All other pairs were scored zero. The segment length for computed is ±3 for charge interactions, and ±2 for hydrophobic interactions (Supplementary Fig. 16). The interaction score was computed at every one residue shift when the above chain shifts laterally from the N- to C-terminus of the neighboring chain. In order to mimic the natural arrangement of Type I collagen, 0.6 D gap (156 amino acids) was set when the interaction scores were calculated (Supplementary Fig. 17). For the PₙXPₙ collagen, (PPG)₁₀ was overlapped to match the full overlap of (PPG)₁₀ in self-assembly process (Supplementary Fig. 16).

## Computational modeling of association of attachment modules

The structural model of Pₙ was obtained by shortening or elongating a high-resolution (1.3 Å) crystal structure of (PPG)₁₀ (PDB: 1K6F)[55]. Pₙ was aligned along the principal *z*-axis by minimizing the projection on the *x*-*y* plane. The relative positions of the other off-centered triple helices in Pₙ + Pₙ were described with five geometric parameters, Ω, θ, φ, *r*, and Δh[31] (Supplementary Fig. 22). Ω was a tilting angle between two triple helices. φ was the self-rotation angle of the off-centered triple helices, which rotated around the centered one with the angle θ. *r* and Δh were the distances along *x*- and *z*-axis, respectively. When Ω = 0°or 180°, association energies of Pₙ + Pₙ were optimized with a Monte Carlo Simulated Annealing (MCSA) program with 16,000 steps from 40 to 0.001 °C. During the optimization, all atoms within the same triple helices were kept fixed in order to keep the intra-helical energies constant. The *van der Waals* energy between the two triple helices was calculated using a 12-6 Lenard−Jones potential with a distance cutoff

6.0 Å. The AMBER ff14SB force field parameters were used for non-bonding interactions[30].

$R_n + E_n$ association was modeled with a similar protocol used for $P_n$. Atomic models were generated by mutating $(PPG)_{10}$ with protCAD. Parallel and antiparallel associations ($\Omega = 0°$, 180°) were sampled. Together with the five geometric parameters, the side chain rotamers of Arg and Glu were optimized with MCSA.

## Simulation of collagen self-assembly

A coarse-grained model of self-assembly simulating the process of diffusion-limited aggregation (DLA) was established with a modified Java script previously reported[36]. The simulation flowchart was presented in Supplementary Fig. 18. A triple helical fragment, $[(PPG)_{10}]_3$, was simplified as a sphere with 1 nm diameter. In $P_{10}BP_{10}$, the adhesive module $P_{10}$ was simplified as 10 spheres. As the helical rise of a triplet with low Pro content is about 0.8–0.9 nm[20], the 27-triplet functional-driver module of $P_{10}BP_{10}$ was modeled as 25 spheres. The spheres formed a rigid rod. A seed rod was placed initially at the center and additional rods were placed at the periphery to randomly diffuse along a 3D hexagonal lattice until contacting an existing rod. Considering viscous resistance, the probability of diffusing axially or radially was adjusted:

$$f = \eta S \frac{dv}{dt} \tag{1}$$

where $f$ is viscous resistance; $\eta$, viscosity of the fluid; $S$, cross-sectional area in the direction of movement; $\frac{dv}{dt}$, fluid velocity gradient.

The probability of a movement $P$ along axial ($P_z$) and radial ($P_{xy}$) is inversely proportional to the resistance.

$$P_z/P_{xy} = f_{xy}/f_z = \eta S_{xy} \frac{dv}{dt} / \eta S_z \frac{dv}{dt} \tag{2}$$

In this simulation, it can be assumed that $\frac{dv}{dt}$ and $\eta$ have the same values in the axial and lateral movements. Collagen can be considered as a cylinder with length $L$ and diameter $D$.

$$P_z/P_{xy} = S_{xy}/S_z = LD/(\pi D^2/4) = 4L/\pi D \tag{3}$$

DLA models of fiber assembly were generated by repeating, 100 times, the simulation of seven sets of rods (50, 100, 300, 600, 1000, 1500, and 3000 rods), where each rod represented the $P_{10}BP_{10}$, $P_{10}BBP_{10}$, or $P_{10}ABCP_{10}$ construct. Fiber length and width were taken to be the weighted average of the total set of simulations per rod type.

## Cell proliferation assay

A 96-well culture plate (3599, Corning, USA) was pre-coated by 10 μg cm$^{-2}$ collagens, including B, $P_5BP_5$, $P_{10}BP_{10}$, $P_{10}BBP_{10}$ and Type I collagen (C7661, SIGMA, USA), at 4 °C overnight. Then the sample solution was removed and the plate was dried in the air for 12 h. After 2 h ultraviolet sterilization, the plate was blocked by 5% BSA (A600332-0100, Sangon Biotech, China) for 2 h. Then 2000 MC-3T3 cells were seeded into the wells and Cell Counting Kit-8 (BA00208, BIOSS, China) was used to detect the cell proliferation according to the manufacturer's instructions. At days 0, 1, 3, and 5, 10 μl Cell Counting Kit-8 solution was added into each well and the absorbance at 450 nm was measured through SYNER GY H1 (BioTek, USA). The culture medium contains 90% α-MEM medium (SH30265.01, Hyclone, USA), 10% fetal bovine serum (FND500, ExCell, China), and 1% Penicillin–Streptomycin (KGY0023, KeyGEN, China). The culture medium was renewed every two days.

## Cell adhesion assay

A 96-well culture plate was coated with the same samples as the cell proliferation assay. The $10^4$ MC-3T3 cells were seeded into each well for 6 h. The culture medium was removed and the plate was washed with PBS gently 3 times. A new culture medium was supplemented and 10 μL Cell Counting Kit-8 solution (BA00208, BIOSS, China) was added to each well. The absorbance at 450 nm was measured through SYNER GY H1 (BioTek, USA). The adhesion percentage was calculated by dividing the number of adherent cells by the seeding cells.

## Quantitative polymerase chain reaction assay (qPCR)

A 96-well culture plate was coated with the same samples as the cell proliferation assay. After cell seeding for 6 h, the adhesion-related gene expression of MC-3T3 cells was detected with qPCR. MC-3T3 cells were also cultured in the plate under osteoblastic differentiation induction for 7 days, and then osteoblastic gene expression was detected with qPCR. Total RNA was isolated from cells using TRIzol Reagent (15596-018, Invitrogen, USA) and cDNA synthesis was conducted using the Transcriptor First Strand cDNA Synthesis Kit (Roche, Switzerland) according to the manufacturer's instruction. The qPCR experiments were performed with an SYBR Premix Ex Taq II kit (Takara, Japan). The gene expression levels were normalized to the *Gapdh* expression level. The sequence of the primers used is listed in Supplementary Table 3.

## Cytoskeleton staining

A 24-well culture plate was coated with the same samples as those in the cell proliferation assay. About 4000 MC-3T3 cells were cultured in each well for 6 h. Then, the plate was washed through PBS 3 times and the cells were fixed by 4% PFA for 30 min. After incubation in 0.3% Triton X-100 (ST797, Beyotime, China) for 15 min, cells were stained with FITC-Phalloidin (KD0382, Kingmorn, China) for 30 min. Finally, cell nuclei were counterstained with DAPI (D9542, Sigma, USA) for 10 min. The images were taken using a fluorescence microscope. The area occupied by cells was determined by Image J[56].

## Alkaline phosphatase staining

A 24-well culture plate was coated with the same sample as those in the cell proliferation assay. MC-3T3 cells were cultured in a plate under osteoblastic differentiation induction for 7 days. The plate was washed with PBS 3 times, and 4% PFA was used to fix cells for 30 min. Then the Alkaline phosphatase (ALP) staining was performed using the ALP Staining Kit (P0321M, Beyotime, China) according to the manufacturer's instructions. The cells were incubated with ALP staining solution at 37 °C for 1 h. The images were taken using a light microscope and the ALP staining area was determined by Image J.

## Tomography data collection and analysis

EM grids for cryoET structural studies were prepared using a self-assembled collagen solution. $P_{10}BP_{10}$ and $P_{10}BBP_{10}$ samples were mixed with 6 nm nanogold particles to facilitate tilt series alignment during image processing. An aliquot of 3.5 μl of collagen fibers was then applied onto glow-discharged Quantifoil holey grids (R2/2, Cu, 200 mesh; Quantifoil) before plunge frozen using a Leica EM GP plunger (Leica Microsystems, Buffalo Grove, IL, USA) in a humidity (95%) and temperature (20 °C) controlled chamber. Images and tilt series of the samples were collected on a Talos Arctica cryo-electron microscope (Thermo Fisher Scientific, Waltham, MA, USA) operated at 200 kV, equipped with a post-column BioQuantum energy filter (the slit was set to 20 eV) and a K2 direct electron detector. Automated data collection was performed using SerialEM[57]. Tomographic data was acquired using a dose-symmetric tilt scheme with 3° intervals and ×31,000 microscope magnification with a corresponding pixel size of 4.36 Å/pixel. Data collection was performed in counting mode, with spot size 9, 100 μm condenser, and 70 μm objective apertures. Tilt series of both $P_{10}BP_{10}$ and $P_{10}BBP_{10}$ were collected at a defocus range

of −8 to −11 µm with a cumulative dose of ~160 $e^-$/Å$^2$. The latest EMAN2 tomography workflow was used for all tilt series alignment and tomogram reconstruction[58].

Quantitative measurements (bandwidths, periodicity, etc.) were performed using functionalities in EMAN2. Volume rendering and visualization of tomograms were done using UCSF Chimera[59]. Measurements of periodicity in real space involved calculating the length of the fiber along multiple periodicities, divided by the total number of periods. To evaluate whether measurements of periodicity in real space v. Fourier space were statistically different, we performed a two-sample *t*-test with equal variance.

## Statistics and reproducibility
All numerical data are expressed as the mean ± s.d. The significance of differences among groups for cell culture were assessed using ordinary one-way ANOVA with multiple comparisons test using Prism 8.2.1 (**** $P < 0.0001$, ** $P < 0.01$, * $P < 0.05$). $P < 0.05$ was considered statistically significant. Experiments were repeated at least three times independently with similar results in all agarose gel electrophoresis, SDS-PAGE, TEM, Cryo-ET, and AFM experiments. The statistical analyses were performed with OriginPro 2019b software for the TEM at $n = 200$, and for the DLA models at $n = 100$.

## Data availability
Electron density maps of P10BP10 and P10BBP10 synthetic collagen fibers have been deposited in the Electron Microscopy Data Bank under the accession codes EMD-27664 and EMD-27665, respectively. Scripts and code are available at https://github.com/JinyuanHu. The data that support the findings of this study are available from the corresponding author.

## Code availability
The protCAD modeling platform used in this study is available at https://github.com/JinyuanHu/protcad/releases. The code for AFM data processing is available under https://github.com/JinyuanHu/FFT_AFM/releases. The code for DLA simulation is available at https://github.com/JinyuanHu/dla/releases.

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

## Acknowledgements

We wish to thank the core imaging facilities at the School of Biotechnology, Jiangnan University, and Robert Wood Johnson Medical School, Rutgers University for help obtaining TEM images. This work was supported by the National Key R&D Program of China (No. 2018YFA0901600) and the National Natural Science Foundation of China (No. 22078129) to FX, National Institute of Health of US (No. R35-GM136431) to JB, NASA NAI 80NSSC18M0093 (VN), and National Natural Science Foundation of China (No. 82270963, No. 82061130222) and Foundation of SCST (No. 20XD1424000, 201409006400) to YS.

## Author contributions

F. X., J. H., and V. N. designed experiments, synthesized materials, performed experiments, analyzed data and wrote the manuscript. J. L. and Y. S. collected and analyzed the cell culture data and wrote the manuscript. J. J. and W. D. collected and analyzed the cryoET data and wrote the manuscript. L. W. collected the TEM data. K. M. took part in DLA simulation and discussions. J. R., and J. B. collected and analyzed the AFM data. F. X. and V. N. supervised the research and wrote the manuscript. All authors discussed the results and commented on the manuscript.

## Competing interests

Patent US20210079064A1—"Preparation of Type I Collagen-Like Fiber and Method for Regulating and Controlling the D-periodic of Fiber Thereof" Fei Xu, Jinyuan Hu, Vikas Nanda, David I. Shreiber, Meng Zhang, Sonal Gahlawat.

## Additional information

**Correspondence and requests** for materials should be addressed to Yao Sun, Vikas Nanda or Fei Xu.

