## [Peer Review File · Nature Communications]

Design of synthetic collagens that assemble into
supramolecular banded fibers as a functional biomaterial
testbedREVIEWER COMMENTS

Reviewer #1 (Remarks to the Author):

This contribution addresses a longstanding and critical need in the collagen field related to the recapitulation of native collagen structures in vitro. The team has presented an interesting approach with fantastic results in terms of collagen assembly using designed sequences. Further, the connection and use of modeling and experimental approaches to achieve the outcomes is also highly relevant and important moving forward in this field. The impact of such approaches over the long term is clear in terms of both fundamental insight into collagen structure-function, as well as potential translational goals. Thus, overall, I am very enthusiastic about the research and its impact on the field moving forward.

I do have some comments and questions that would help clarify points in the work and hopefully improve impact. I would add that none of these points is a major concern, they are more minor issues and points to clarify.

Model of Assembly - A clearer drawing/cartoon or model of: (a) triple helical assembly, I still find this difficult to clarify with the explanations provided and the available Figures – in other words, what assures triple helix formation instead of dimers or other multimers, (b) what regulates assembly of the helices into higher order structures in terms of limiting the final diameters of the structures in the images provided, as many of these seem to reach a similar limit – which is also different than what is found in the natural collagen-based ECMs.

Mechanics – There is an absence of mention of mechanical properties throughout the paper. I find this odd, as collagen assembly in and of itself is key from a fundamental perspective, however, ultimately, we do not need this if we just want to make collagen biomaterial matrices to grow cells – there are companies to already generate gelatins from many sources including heterologous host systems. The need is higher order assembly to match the remarkable mechanical (and biological) behavior of collagen-based tissues. Thus, some integration of mechanics into the work would seem essential – or perhaps some AFM related measurements of the assemblies to confirm outcomes.

2D vs 3D – What is the evidence to confirm that these are 3D assemblies and not just 2D assemblies. The images are fantastic and either way, the work is impactful, but ultimately towards the stated goal of the study, some clear evidence of 3D would help – e.g., perhaps cross sections of the assemblies and images?

Cell Outcomes – I would be more cautious in the conclusions from the cell work. First, differences in molecular weight and density of assembly can directly impact cell responses. Second, there are no mature markers of bone tissue formation mentioned (e.g., calcium deposition). Without this, I would be more inclined to emphasize cell compatibility and related themes, and not to over compare outcomes with traditional collagen in terms of osteogenesis.

Stability – It is not clear how stable these assembled collagens are over time under different conditions. How do time, pH and temperature impact the retention of structures. These are again points to allow better comparison to native collagens, to see how similar or not these new materials behave. There is nice data on how assembly under different conditions provides differences in outcomes, but the disassembly/stability is not clear.

Reviewer #2 (Remarks to the Author):

In this manuscript Hu et al use a clever strategy to design synthetic collagens that form banded fibres akin those observed in natural collagens. They do so by using a “functional driver”, a fragment of collagen-like bacterial protein, flanked by “sticky” ends. These sticky ends were designed either as hydrophobic stretches, or stretches of charged residues (+ on one end and - on the other end). The authors show that such a system can not only assemble into banded fibres, whose band width can be tuned by changing the length of the functional region, but that they also

support proliferation, adhesion, and osteogenic differentiation of osteoblast precursor cells, which I found very nice. The authors also use armistice modelling to calculate and explain the critical length of the "sticky" ends needed for the stable fibre assembly, and they use coarse-grained simulations to recapitulate assembly in a model.

Overall, I found the paper very nice. The general idea of using "sticky ends" flanking neutral regions to form banded fibrils is not new, and has been used before in shorter triple peptides, see eg Ref 19. The dynamic assembly of such molecules has also been explored in simulations (see <https://doi.org/10.1016/j.bpj.2020.09.013>, which should be added to the list of References), and in calculations for native collagen (eg Ref 32). However, this paper manages to go beyond the previous efforts by i) cleverly using the functional driver, which also makes the system compatible with living cells and ii) achieving banding of ~60nm, which is similar to those found in natural collagen (previous synthetic collagen-like peptides typically achieved only shorter bandings). The extension to exploring of the biological compatibility of such molecules I also found very compelling, and the paper shows a true interdisciplinary effort by combining important aspects of biochemistry, cellular biology, and physical and physicochemical modelling.

I am happy to support publication in Nature Comm. I suggest addressing the following comments:

- The authors compare the functionality of molecules that do not assemble into fibres versus those that assemble into fibres. However, it seems to me that the finer banding does not play a role, i.e. P10BP10 (~25nm dark band) and P10BBP10 (~50nm dark band) seem to perform similarly well. Can the authors comment? Is the banding even important for this functionality, or are any fibrils that have collagen-like properties sufficient? I think this is an important point to clarify and comment on in the paper.

- I found the modelling part, Fig 3 and SI Fig 12&13, generally very nice. However, I am confused with why the measurement of the fibril periodicity is not shown, along with the distribution of the measured periodicities, given that this is the central point of the paper. Can that be added?

-The measurements of fibril thickness and length in Fig 3G I found a bit misleading, as I would expect this to depend on the used effective concentration/ number of rods used, and the effective dimensions of the "assembly volume" in the DLA model. So the fact that the lengths and thicknesses matches the experiment I did not find very strong. Possibly the aspect ratio is a better indicator of the agreement.

- I am confused to why the functional region does not give staggered assembly on its own (Extended Data Fig 12) if it's a collagen-like protein. Can the authors explain? What makes it collagen-like?

- The term "synthetic collagens" could be misleading, as these molecules were not synthesised bottom-up. This of course depends on the community that a reader belongs to, but the authors might wish to clarify this earlier on in the paper.

-The authors mention several times that these molecules could be safer to use as biomaterials, as they are not of animal origin. But they still involve using E. Coli; is this obviously safer?

Reviewer #3 (Remarks to the Author):

Strengths:

The study by Hu et al. presents a set of synthetic proteins that incorporate both natural (bacterial-collagen-derived) and designer repetitive sequence fragments to prepare collagen-like banded nanofibers (CLBNs). The design of the proteins is elegantly simple, and the microscopy data (especially dry TEM images) are striking. Further development of the technology may yield synthetic collagen-mimicking fibrous biomaterials with low batch-to-batch variability and programmable functionality.

Designing banded nanofibers is quite difficult with short collagen-like peptides prepared by solid-phase peptide synthesis (<https://pubs.acs.org/doi/10.1021/jacs.5b03326>; <https://pubs.acs.org/doi/10.1021/ja504377s>; <https://www.nature.com/articles/nchem.2556>), although there is an early report of a D-periodic banded collagen nanofiber based on a triblock design (<https://pubs.acs.org/doi/10.1021/ja0758990>). The proposed design strategy by Hu et al. obviates this issue by expressing long (≥ 300 residues) proteins in *E. coli*, with the limitation that there are no hydroxyproline (O) residues in the designer proteins. Hydroxyproline residues are present in many ECM-binding domains in mammalian collagen (e.g., GFOGER), and its absence may limit the biomedical application of CLBNs (such as hosting mutations of interest relevant to osteogenesis imperfecta).

Despite key limitations, I believe that the current manuscript may substantially advance the frontier of the field, after the authors address the following questions and suggestions.

Major Issues and Remaining Questions:

1. The authors claim that "We also experimentally demonstrate that our synthetic fibers stimulate osteogenesis". This is a strong statement that is not adequately supported by their data and analysis. It is not clear whether the role of the collagen-like proteins is minimal, while osteogenic differentiation media does the heavy lifting of inducing differentiation in MC-3T3 cells in vitro. Osteogenesis is a multi-step complex process, and if the authors want to assert that the CLBNs stimulate osteogenesis, they should demonstrate it in an animal model (for example, see <https://www.pnas.org/doi/full/10.1073/pnas.1813000115>). In addition, they may incubate CLBNs in presence of hydroxyapatite nanocrystals to mechanistically probe whether the nanofibers can enable biomineralization (please see the following model: <https://www.nature.com/articles/ncomms2720>). If the hydroxyapatite nanocrystals intercalate within the gap region of CLBNs or are templated by the nanofibers, the authors should be able to study it using TEM and electron diffraction.
2. The hierarchical self-assembly model is not entirely clear in the manuscript. Are the triple helices parallel or anti-parallel in the assembly? Do the ends of the triple helices interact with each other? In the CD plots, do the thermal transitions indicate both CLBN unraveling and triple helical melting (Extended Data Fig. 6). Is it possible that the banded nanofibers disintegrate before the triple helical melting (in some CLBNs, there seem to be two melting transitions, similar to <https://www.nature.com/articles/nchem.1123>)? If so, it would complicate the in vivo application of CLBNs in a physiologically relevant temperature (37°C).
3. What are the inter-triple helical distances in the CLBNs? Fiber diffraction may help answer this question. "Calculation of Inter-Triple helical Interactions" is also a bit crude. What is the basis of assigning charged pairs (K/R, D/E) and hydrophobic pairs (V, M, I, L, F, P) the same score? Charge-pair interactions are directional, which is not adequately considered.
4. What are the material properties of the CLBN materials? Why don't the long nanofibers entangle to form hydrogels (similar to denatured collagen I)?
5. The TEM images in Figure 2 and Extended Data Figures 8–11 are great. However, they all potentially suffer from drying and aggregation artifacts, and they may be affected by the electrostatic interaction with polytungstate anions. AFM (Extended Data Fig. 5) does not have the stain-artifact issues, but the image does not clearly show nanofibers; a liquid-crystalline arrangement may yield the same result as Extended Data Fig. 5. Potentially, vitreous-ice cryo-TEM may help authors experimentally explore the structural features of self-assembly with more confidence, supporting their diffusion limited aggregation model.

Minor Points:

1. Some of the references are incomplete.
2. I urge the authors to discuss the limitations of the study and scope for further advances at the

end of the Discussion section—this will help the readers gain a balanced perspective.

REPOSE TO REVIEWERS' COMMENTS

Reviewer 1:

1.1 (a) Model of Assembly - A clearer drawing/cartoon or model of: (a) triple helical assembly, I still find this difficult to clarify with the explanations provided and the available Figures – in other words, what assures triple helix formation instead of dimers or other multimers,

The folding of three collagen polypeptide chains into the triple-helix has been extensively studied by other groups (see reference 51). In order to specifically form a trimer – the sequence must contain an extended series of [glycine – xxx – yyy]_n repeats. Since identifying this sequence motif is not the innovation in our designs, we do not discuss this aspect in detail in Fig. 1. However, we appreciate the reviewer's concern and have added a sentence to the legend pointing readers to a comprehensive review article on collagen folding and assembly (reference 51).

1.1 (b) what regulates assembly of the helices into higher order structures in terms of limiting the final diameters of the structures in the images provided, as many of these seem to reach a similar limit – which is also different than what is found in the natural collagen-based ECMs.

There are multiple processes that contribute to the final diameters of supramolecular fibrous structures, such as those formed by collagen. A thermodynamic contribution is the relative surface energy that can change as the fiber grows, modulating the balance between a solvated monomer and one bound to the growing fiber. A kinetic factor is the role of concentration and diffusion in fiber growth. We have focused on this second aspect in our diffusion-limited aggregation simulations, where we show that the aspect ratio – the ratio of length to diameter – can be explained largely by diffusion kinetics.

1.2 Mechanics – There is an absence of mention of mechanical properties throughout the paper. I find this odd, as collagen assembly in and of itself is key from a fundamental perspective, however, ultimately, we do not need this if we just want to make collagen biomaterial matrices to grow cells – there are companies to already generate gelatins from many sources including heterologous host systems. The need is higher order assembly to match the remarkable mechanical (and biological) behavior of collagen-based tissues. Thus, some integration of mechanics into the work would seem essential – or perhaps some AFM related measurements of the assemblies to confirm outcomes.

While this work focuses on design of a banded fiber morphology, we certainly agree that mechanical properties of collagen materials are central to their function. In order to address mechanics, we have now performed micro-rheology experiments using a particle tracking method we previously applied to collagen hydrogels. We prepared P₁₀BP₁₀ at a low (1mg/mL) and high (5mg/mL) concentrations. Unlike Type I collagen, P₁₀BP₁₀ does not form a viscoelastic gel, while natural Type I collagen forms gels at the same concentrations. Therefore, the mechanical properties of synthetic and natural collagens are not comparable.

We have include **Supplementary Fig. 7** describing this experiment:

Supplementary Fig. 7. Microrheology storage and loss moduli, G' and G'' , and tilting tube test of the type I collagen (a, b) and P₁₀BP₁₀ (c, d) solutions at 1 and 5 mg/mL in 10 mM phosphate buffer (pH 7.4).

We described the results of the micro-rheology experiments in the manuscript as below:
 “Although P₁₀BP₁₀ assembles into fibers, micro-rheology experiments show fibers do not form a viscoelastic gel either at a low (1 mg/mL) or high (5 mg/mL) concentration (Supplementary Fig. 7).”

Although the synthetic collagens do not form hydrogels, they still may have potential as coating materials on surfaces such as metal implants (Rammelt, Illert et al. 2006). Additionally, there likely are ways to tune the sequence of the functional-driver module to mediate hydrogelation and develop viscoelastic behavior similar to natural biomaterials. We include this in the Discussion: “Although the designed collagens did not form hydrogels, they still can be potentially used as coating materials on metal implants⁵⁰. It may be possible to design functional-drivers that promote formation of a supramolecular viscoelastic gel.”

1.3 2D vs 3D – What is the evidence to confirm that these are 3D assemblies and not just 2D assemblies. The images are fantastic and either way, the work is impactful, but ultimately towards the stated goal of the study, some clear evidence of 3D would help – e.g., perhaps cross sections of the assemblies and images?

This is an important question raised by several reviewers. We characterized the fibers with CryoET. The results conclusively show the fibers are three-dimensional. We have added a new figure (Fig. 2) into the main text as below:

Figure 2. Cryo-electron tomography of P₁₀BP₁₀ collagen fibrils. **a** Tilt series of collagen fibers at tilt angles ranging from -60° to +60°. **b** Slice views of the tomogram of P₁₀BP₁₀ collagen fibers with characteristic banding pattern. Inset in (b) shows a zoomed-in view with measurement of periodicity. **c** Cross-section of a collagen bundle as indicated by a dashed yellow line in (b). **d** Fourier transform of the image in (b) showing banding periodicity. **e** Volume rendering of collagen fibers showing dense overlap and sparse gap zones.

A similar result for P₁₀BBP₁₀ showing the longer dimension of the functional-driver, while maintaining 3D assembly is shown in Supplementary Fig. 22:

Supplementary Figure 22: Cryo-electron tomography of P₁₀BBP₁₀ collagen fibrils. **a** Tilt series of collagen fibers at tilt angles ranging from -60° to +60°. **b** Slice views of the tomogram of P₁₀BBP₁₀ collagen fibers with characteristic banding pattern. Inset in (b) shows a zoomed-in view with measurement of periodicity. **c** Cross-section of a collagen bundle as indicated by a dashed yellow line in (b). **d** Fourier transform of the image in (b) showing banding periodicity. **e** volume rendering.

1.4 Cell Outcomes – I would be more cautious in the conclusions from the cell work. First, differences in molecular weight and density of assembly can directly impact cell responses. Second, there are no mature markers of bone tissue formation mentioned (e.g., calcium

deposition). Without this, I would be more inclined to emphasize cell compatibility and related themes, and not to over compare outcomes with traditional collagen in terms of osteogenesis.

We appreciate the reviewer's distinction between osteogenesis and the preliminary stages of osteoblast formation. The latter process is directly reported by markers we used. To accurately reflect this, we refer to the cell outcomes as "osteoblastic differentiation".

1.5 Stability – It is not clear how stable these assembled collagens are over time under different conditions. How do time, pH and temperature impact the retention of structures. These are again points to allow better comparison to native collagens, to see how similar or not these new materials behave. There is nice data on how assembly under different conditions provides differences in outcomes, but the disassembly/stability is not clear.

To determine stability as a function of temperature, time and pH as requested, we performed CD wavelength scans and heat denaturation profiles of P₁₀BP₁₀ under various pH from 3 to 11. The MRE signals and melting temperatures were similar, indicating that the triple helices were stable under various conditions. We also incubated P₁₀BP₁₀ at 4 °C at pH from 3 to 11, and from 12 hours to 70 days at pH 7. Morphology of the fibers were very similar. In the Results section, we have noted the stability of the fibers. "Assembly of banded fibers was robust, and reproducible multiple times under a variety of solution conditions including 50 to 150 mM NaCl concentrations and pH 3-11, incubation time from 0.5 to 70 days, and temperatures from 4 to 37 °C (Supplementary Fig. 4-6)."

We present the primary data in the Supplementary Fig. 4 (temperature), 5 (time) and 6 (pH).

Supplementary Fig. 4. Transmission Electronic Microscopy (TEM) images of P₁₀BP₁₀ fibers at various temperatures and salt concentrations. The samples were prepared with 0.5 mM concentration in 10 mM phosphate buffer at pH 7 and incubated at 4°C (**a-c**), 20°C (**d**), 30°C (**e**), and 37°C (**f**) for 3 days. The samples were also prepared under the same condition as Fig. 1f, in a 10 mM phosphate buffer at pH 7 with 50 mM NaCl (**g**), 100 mM NaCl (**h**), and 150 mM (**i**). Before TEM imaging, the samples were negatively stained with 0.75% phosphotungstic acid.

Supplementary Fig. 5. Transmission Electronic Microscopy (TEM) images of P₁₀BP₁₀ fibers self-assembled at various time. The samples prepared under the same condition as Fig. 1f, in a 10 mM phosphate buffer at pH 7 for 0.5 days (a), 2.5 days (b), 4.5 days (c), 7.5 days (d), 35 days (e), 70 days (f).

Supplementary Fig. 6. Circular dichroism characterization and Transmission Electronic Microscopy (TEM) images of P₁₀BP₁₀ fibers self-assembled at different pH. Wavelength scans (a), thermal transitions curves (b), and one order derivative of thermal transitions (c) of the samples. The samples prepared under the same condition as Fig. 1f, in a 10 mM phosphate buffer at pH 3 (d, e), pH 5 (f, g), pH 9 (h, i), and pH 11 (j, k), for 3 days.

Reviewer 2:

2.1 Overall, I found the paper very nice. The general idea of using “sticky ends” flanking neutral regions to form banded fibrils is not new, and has been used before in shorter triple peptides, see eg Ref 19. The dynamic assembly of such molecules has also been explored in simulations (see <https://doi.org/10.1016/j.bpj.2020.09.013>, which should be added to the list of References), and in calculations for native collagen (eg Ref 32). However, this paper manages to go beyond the previous efforts by i) cleverly using the functional driver, which also makes the system compatible with living cells and ii) achieving banding of ~60nm, which is similar to those found in natural collagen (previous synthetic collagen-like peptides typically achieved

only shorter bandings). The extension to exploring of the biological compatibility of such molecules I also found very compelling, and the paper shows a true interdisciplinary effort by combining important aspects of biochemistry, cellular biology, and physical and physicochemical modelling. I am happy to support publication in Nature Comm. I suggest addressing the following comments:

We added the reference by Hafner *et al.* about the simulation of Type I collagen assembly in the main text. We thank the reviewer for the reference and hope to use the model of elastic rods in future work.

2.2 - The authors compare the functionality of molecules that do not assemble into fibres versus those that assemble into fibres. However, it seems to me that the finer banding does not play a role, i.e. P10BP10 (~25nm dark band) and P10BBP10 (~50nm dark band) seem to perform similarly well. Can the authors comment? Is the banding even important for this functionality, or are any fibrils that have collagen-like properties sufficient? I think this is an important point to clarify and comment on in the paper.

A key feature of banded fibers is that they align receptor sites in close lateral proximity (within 10-20 Å). This proximity of binding sites could promote the clustering of integrin receptors, resulting in focal adhesions that would stimulate downstream cytoskeletal assembly. We include this description in the text and have added a schematic figure (Supplemental Fig. 23):

Supplementary Fig. 23: Hypothesized mechanism for banded synthetic collagen inducing osteoblastic differentiation. Both P10BP10 and P10BBP10 laterally align multiple GFPGER sites along the fiber axis. At each position, multiple integrins can bind, resulting in receptor clustering. This activates downstream signaling through factors such as talin and vinculin, and eventually driving cytoskeletal changes.

2.3 - I found the modelling part, Fig 3 and SI Fig 12&13, generally very nice. However, I am confused with why the measurement of the fibril periodicity is not shown, along with the distribution of the measured periodicities, given that this is the central point of the paper. Can that be added?

The periodicity is strictly enforced in the DLA model by the distance between adhesive modules in the coarse-grained model. As such, this is not a measurable property. The goal of these calculations was to model width and length of supramolecular clusters of collagens.

2.4 -The measurements of fibril thickness and length in Fig 3G I found a bit misleading, as I would expect this to depend on the used effective concentration/ number of rods used, and the effective dimensions of the “assembly volume” in the DLA model. So the fact that the lengths and thicknesses matches the experiment I did not find very strong. Possibly the aspect ratio is a better indicator of the agreement.

We now specify the predicted number of the rods that match widths/lengths observed in the TEM in the legend of Fig. 4: “The assembly process was simulated by allowing **1200** rods to diffuse randomly along a hexagonal lattice, until contacting an existing rod through their adhesive ends.”

As the reviewer suggests, aspect ratio is a dimensionless quantity that could potentially be independent of the number of molecules. We have included this analysis in the revision. As the numbers of rods increased in the simulation, the thickness and length increased but length-thickness ratio decreased only slightly (Supplementary Fig. 19 and 20). We displayed the distribution of the aspect ratios extracted from TEM images (Supplementary Fig. 20).

Supplementary Fig. 19. Thickness and Length of the fibers simulated with Diffusion-Limited Aggregation (DLA) of the synthetic collagens, P₁₀XP₁₀ (X=B, BB, ABC).

Supplementary Fig. 20. Comparison of length:thickness ratio of the fibers in DLA simulation vs. the TEM images (analyzed using ImageJ). (a) Length:thickness ratio was plotted against numbers of chains in the DLA simulation. Dashed lines indicated the average length:thickness ratios from the TEM images (analyzed using ImageJ). (b) The distribution histogram of length:thickness ratios from the TEM images.

We also modified the main text accordingly as below:

Page 12, Paragraph 3, and Line 2

“Simulated fibers were generated for P₁₀BP₁₀ containing 50 to 3000 individual triple-helices. The length and thickness of the simulated fibers increased with the number of rods (Supplementary Fig. 19). However, the aspect ratio (length/thickness) decreased only slightly, still close to the values extracted from TEM (Supplementary Fig. 20), consistent with previous diffusion models of collagen (Fry, Mohammad et al. 2004). Simulated dimensions best matched observations for fibers composed of ~1200 rods (Fig. 4g).”

2.5- I am confused to why the functional region does not give staggered assembly on its own (Supplementary Fig 12) if it's a collagen-like protein. Can the authors explain? What makes it collagen-like?

A previous study showed that Scl2 protein did not form fibers on its own (Yoshizumi, Yu et al. 2009). Consistently, the fragment B, which was the central part of Scl2 protein, did not form fibers, which was taken as a negative control compared to the positive sample, the fiber-forming P₁₀BP₁₀.

2.6 - The term “synthetic collagens” could be misleading, as these molecules were not synthesised bottom-up. This of course depends on the community that a reader belongs to, but the authors might wish to clarify this earlier on in the paper.

We hope the meaning term ‘synthetic’ will be clear in the context of this study. We consistently contrast synthetic collagen to ‘natural’ Type I collagen. It is synthetic in the sense these proteins are not only man-made, but also not found in nature. While the functional drivers are derived from fragments of natural proteins such as Scl2, we have altered their sequence, and they are not being used in a context relevant to their biological function. We had considered other typologies such as ‘artificial’ and ‘designed’ collagens, but we do not feel that these are more precise than ‘synthetic’

2.7 -The authors mention several times that these molecules could be safer to use as biomaterials, as they are not of animal origin. But they still involve using E. Coli; is this obviously safer?

An *E. coli* expression system is usually the first choice to produce recombinant proteins for laboratory investigation, initial development of commercial application, and a benchmark for comparison among various systems (Chen 2012). Animal protein sources can contain contaminants that can promote disease in humans such as prions, bacteria or viruses. Recombinant protein production methods allow for purer more consistent material, a critical component of safety. Once a protein has proved to have desired properties, it is quite feasible to produce the protein with other expression systems, such as *Corynebacterium glutamicum*, which is safe to human (Joo-Young, Yoon-Ah et al. 2016).

Reviewer 3:

3.1. The authors claim that "We also experimentally demonstrate that our synthetic fibers stimulate osteogenesis". This is a strong statement that is not adequately supported by their data and analysis. It is not clear whether the role of the collagen-like proteins is minimal, while osteogenic differentiation media does the heavy lifting of inducing differentiation in MC-3T3 cells in vitro. Osteogenesis is a multi-step complex process, and if the authors want to assert that the CLBNs stimulate osteogenesis, they should demonstrate it in an animal model (for example, see <https://www.pnas.org/doi/full/10.1073/pnas.1813000115>). In addition, they may incubate CLBNs in presence of hydroxyapatite nanocrystals to mechanistically probe whether the nanofibers can enable biomineralization (please see the following model: <https://www.nature.com/articles/ncomms2720>). If the hydroxyapatite nanocrystals intercalate within the gap region of CLBNs or are templated by the nanofibers, the authors should be able to study it using TEM and electron diffraction.

We agree that the current work only reports on the preliminary stages of osteoblast differentiation, and have adjusted our claims accordingly.

3.2.a The hierarchical self-assembly model is not entirely clear in the manuscript. Are the triple helices parallel or anti-parallel in the assembly? Do the ends of the triple helices interact with each other?

We do not know if the interactions between proline-rich adhesive domains are parallel or antiparallel (or mixed) as any of these scenarios would result in the same banding morphology. Molecular modeling suggests that the antiparallel association would be favored (Supplementary Figure 21d). For the R-E adhesive domains, a parallel association would be consistent with the model. The cryo-ET tomograms are not sufficient resolution to answer this questions.

3.2.b. In the CD plots, do the thermal transitions indicate both CLBN unraveling and triple helical melting (Supplementary Fig. 6). Is it possible that the banded nanofibers disintegrate before the triple helical melting (in some CLBNs, there seem to be two melting transitions, similar to <https://www.nature.com/articles/nchem.1123>)? If so, it would complicate the in vivo application of CLBNs in a physiologically relevant temperature (37°C).

To address the reviewer's question and in response to a similar concern (see comment 1.5), we incubated the fiber at 37 °C for three day. Fiber morphology remained unchanged as assessed by TEM (Supplementary Fig. 5 (f)).

We also performed Differential Scanning Calorimetry (DSC) on synthetic collagens P₈BP₈, P₁₀BP₁₀, P₁₀BBP₁₀. Two peaks in DSC were consistent with two transitions in the CD melting curves (Supplementary Fig. 10). The narrow and sharp shape of the major peaks was resembled to those of bacteria collagen-like proteins, from which the functional-driver modules were derived, in previous study (Yoshizumi, Yu et al. 2009, Hu, Wang et al. 2021). The major peaks of the synthetic collagens were located between 50-60 °C, higher than those of the collagen-like proteins located between 30-40 °C (Yoshizumi, Yu et al. 2009, Hu, Wang et al. 2021). The increase in the melting temperatures could result from the flanking Pro-rich adhesive modules to the two termini and/or formation of fibers. We would expect the fibers to be stable under physiological conditions. We have included a discussion of the DSC results and their relevance to stability under physiological conditions in both the manuscript and supplementary section.

Supplementary Fig. 9. Differential scanning calorimetry of the synthetic collagen P₈BP₈ (a), P₁₂BP₁₂ (b), P₁₀BP₁₀ (b), and P₁₀BBP₁₀ (d) at 5 mg/mL in 10 mM phosphate buffer (pH 7.4).

3.3.a. What are the inter-triple helical distances in the CLBNS? Fiber diffraction may help answer this question.

Calculated inter-helical distances from atomistic simulations are 11.8 Å for proline-based adhesive domains and 13.3 Å for charged-pair adhesive domains. We did not do fiber diffraction in the current work, but previous powder-diffraction XRD measurements from our lab showed interhelical distances of 11.6 Å for peptides with the proline adhesive domain sequence (see <https://doi.org/10.1021/ja4106545>).

3.3.b. "Calculation of Inter-Triple helical Interactions" is also a bit crude. What is the basis of assigning charged pairs (K/R, D/E) and hydrophobic pairs (V, M, I, L, F, P) the same score? Charge-pair interactions are directional, which is not adequately considered.

We agree that this is a significantly simplified model. It has been used previously for natural collagen (Hulmes & Miller Nature 1979) and recently for another synthetic collagen design (Strawn et. al. Biopolymers 2018). In both cases, this model was able to recapitulate the length of the D-period of banding. That was our motivation in applying this model for our current designs.

3.4. What are the material properties of the CLBN materials? Why don't the long nanofibers entangle to form hydrogels (similar to denatured collagen I)?

We made the micro-rheology experiments, as discussed in response to the comment 1.2. The micro-rheology experiments showed that the fibers of the designed collagens did not form a gel. Although the designed collagens did not form hydrogels, they still can be potentially used as coating materials on metal implants (Rammelt, Illert et al. 2006). In TEM images, the fibers appeared to be randomly clustered rather than entangled with each other. In denatured Type I collagen, unfolded regions from different molecules could be re-annealed and form short stretches of inter-molecular triple helices.

3.5. The TEM images in Figure 2 and Supplementary Figures 8–11 are great. However, they all potentially suffer from drying and aggregation artifacts, and they may be affected by the electrostatic interaction with polytungstate anions. AFM (Supplementary Fig. 5) does not have the stain-artifact issues, but the image does not clearly show nanofibers; a liquid-crystalline arrangement may yield the same result as Supplementary Fig. 5. Potentially, vitreous-ice cryo-TEM may help authors experimentally explore the structural features of self-assembly with more confidence, supporting their diffusion limited aggregation model.

We characterized the fibers with cryo-electron tomography (CryoET). Please refer Response to Comment 1.3 for details.

3.6 Minor Points:

1. Some of the references are incomplete.

The references have been fixed.

2. I urge the authors to discuss the limitations of the study and scope for further advances at the end of the Discussion section—this will help the readers gain a balanced perspective.

We appreciate this comment and have made multiple edits to the manuscript to incorporate the limitations of the current designs.

Response Letter References:

- Chen, R. (2012). "Bacterial expression systems for recombinant protein production: E. coli and beyond." *Biotechnology Advances* **30**(5): 1102-1107.
- Emsley, J., et al. (2000). "Structural basis of collagen recognition by integrin alpha2beta1." *Cell* **101**(1): 47-56.
- Fry, D., et al. (2004). "Cluster Shape Anisotropy in Irreversibly Aggregating Particulate Systems." *Langmuir the Acs Journal of Surfaces & Colloids* **20**(18): 7871.
- Hafner, A. E., et al. (2020). "Modeling Fibrillogenesis of Collagen-Mimetic Molecules." *Biophysical Journal* **119**(9): 1791-1799.
- Hu, J., et al. (2021). "Design Strategies to Tune the Structural and Mechanical Properties of Synthetic Collagen Hydrogels." *Biomacromolecules* **22**(8): 3440-3450.
- Jokinen, J., et al. (2004). "Integrin-mediated Cell Adhesion to Type I Collagen Fibrils." *Journal of Biological Chemistry* **279**(30): 31956-31963.
- Joo-Young, L., et al. (2016). "The Actinobacterium *Corynebacterium glutamicum*, an Industrial Workhorse." *Journal of Microbiology and Biotechnology* **26**(5): 807-822.
- McGuinness, K., et al. (2014). "Morphological diversity and polymorphism of self-assembling collagen peptides controlled by length of hydrophobic domains." *Acs Nano* **8**(12): 12514-12523.
- Parkinson, J., et al. (1995). "Simple physical model of collagen fibrillogenesis based on diffusion limited aggregation." *Journal of Molecular Biology* **247**(4): 823-831.
- Rammelt, S., et al. (2006). "Coating of titanium implants with collagen, RGD peptide and chondroitin sulfate." *Biomaterials* **27**(32): 5561-5571.
- Yoshizumi, A., et al. (2009). "Self-association of streptococcus pyogenes collagen-like constructs into higher order structures." *Protein Science* **18**(6): 1241-1251.

REVIEWERS' COMMENTS

Reviewer #2 (Remarks to the Author):

The authors have generally answered most of my questions and I'm happy to recommend publication.

Re the Q 2.3: while the periodicity is trivially encoded by the design, the mass/density profiles along the fibril could be shown, as these show the width and length relationship of fibrils, along with the gap and overlap regions.

Two more minor comments that I believe should be considered if the authors wish their work to be reproducible by theorists:

- Supplementary fig 16 b the graphical representation should be improved. First of all, no gap region is visible, even though it is claimed to be used. Secondly, the second molecule in the lower row/layer has no left adhesive module next to the driver module.

- Supplementary fig 17 b shows the interaction score type "Pro", but this calculation is nowhere explained. (c) shows attractive ion pairs, but what about repulsive ion pairs? Especially in the case of 0 stagger, there should be many repulsive pairs with score -1 but in the graph, the interaction score has a maximum.

Reviewer #3 (Remarks to the Author):

I congratulate the authors on the depth and clarity of the resubmitted manuscript. Especially, the cryo-electron tomography work was a tour de force. I am happy to recommend the publication of this paper in Nature Communications.

RESPONSE TO REVIEWERS' COMMENTS

Reviewer #2

Re the Q 2.3: while the periodicity is trivially encoded by the design, the mass/density profiles along the fibril could be shown, as these show the width and length relationship of fibrils, along with the gap and overlap regions.

We have added Supplementary Fig. 19 in the revision showing the mass and density profiles of a typical simulated fiber, clearly showing the gap and overlap regions.

Two more minor comments that I believe should be considered if the authors wish their work to be reproducible by theorists:

- Supplementary fig 16 b the graphical representation should be improved. First of all, no gap region is visible, even though it is claimed to be used. Secondly, the second molecule in the lower row/layer has no left adhesive module next to the driver module.

The gap region in synthetic collagen is evident only when several units have assembled (as shown in the response to the previous comment). The goal of this calculation was to assess the association of two units at various residue-level staggers. In this calculation, two adhesive modules would be overlapping. We understand that this might be confusing and have clarified our representation in Supplementary Figure 16b in the legend: "Note than only one adhesive model is shown between adjacent functional drivers to simulate the overlap of these regions in an assembling fiber."

- Supplementary fig 17 b shows the interaction score type "Pro", but this calculation is nowhere explained. (c) shows attractive ion pairs, but what about repulsive ion pairs? Especially in the case of 0 stagger, there should be many repulsive pairs with score -1 but in the graph, the interaction score has a maximum.

This is correct. At zero stagger and optimal staggers thereon, attractive and repulsive interactions are maximal. To clearly show this, we now present a complete breakdown of the interaction energy in Figure 17 into hydrophobic, proline, attractive and repulsive electrostatic interactions.